# Expert Routing with Synthetic Data for Domain Incremental Learning

**Yewon Byun**[1], **Sanket Vaibhav Mehta**[1], **Saurabh Garg**[1], **Emma Strubell**[1], **Michael Oberst**[2],
**Bryan Wilder**[1], **Zachary C. Lipton**[1]

[1]Carnegie Mellon University    [2]Johns Hopkins University

**Reviewed on OpenReview:** `https://openreview.net/forum?id=QdQVfdXnsG`

## Abstract

In many real-world settings, regulations and economic incentives permit the sharing of models but not data across institutional boundaries. In such scenarios, practitioners might hope to adapt models to new domains, without losing performance on previous domains (so-called catastrophic forgetting). While any single model may struggle to achieve this goal, learning an ensemble of domain-specific experts offers the potential to adapt more closely to each individual institution. However, a core challenge in this context is determining which expert to deploy at test time. In this paper, we propose Generate to Discriminate (G2D), a domain-incremental learning method that leverages synthetic data to train a domain-discriminator that routes samples at inference time to the appropriate expert. Surprisingly, we find that leveraging synthetic data in this capacity is more effective than using the samples to *directly* train the downstream classifier (the more common approach to leveraging synthetic data in the lifelong learning literature). We observe that G2D outperforms competitive domain-incremental learning methods on tasks in both vision and language modalities, providing a new perspective on the use of synthetic data in the lifelong learning literature.

## 1 Introduction

To deploy machine learning reliably, it is important to develop methods for adapting models as we encounter environments sequentially in the wild. In medical imaging and risk prediction tasks, practitioners often apply models trained on one hospital's data to make predictions on samples from other institutions (Zech et al., 2018; Pooch et al., 2020; Guan & Liu, 2021). Autonomous vehicle navigation systems must safely operate on different terrains, in different states, and even different countries. Moreover as we adapt to each new environment, it is desirable (whenever possible) that this adaptation comes without loss of accuracy in previously seen environments.

In such domain-incremental continual learning problems, some of the most effective methods rely on a rehearsal buffer to re-train on a portion of past samples (Chaudhry et al., 2019b; Lopez-Paz & Ranzato, 2017). However, these solutions are often not viable in real-world prediction settings, where the landscape of regulatory and industry norms place stringent constraints on sharing data across institutional boundaries. We focus on the domain-incremental learning setting where the set of classes is fixed across domains and explicit data sharing across domains is prohibited. Crucially, domain identifiers are not given during inference time. The goal, after each round, is to produce a system that performs well on test examples drawn at random among all previously seen domains. At any round, our model only has access to the current data and, thus, simply performing gradient updates is liable to cause *catastrophic forgetting*, where performance decays on previously seen domains, even when the tasks are not fundamentally in conflict (McCloskey & Cohen, 1989; French, 1999). To enable lifelong learning in scenarios characterized by stringent constraints on data sharing, several settings for this problem have been proposed (Sodhani et al., 2022). For example, *generative replay* methods utilize generative models to create synthetic samples for experience rehearsal (Shin et al., 2017; Sun et al., 2020; Qin & Joty, 2022). However, these approaches have largely under-performed state-of-the-art

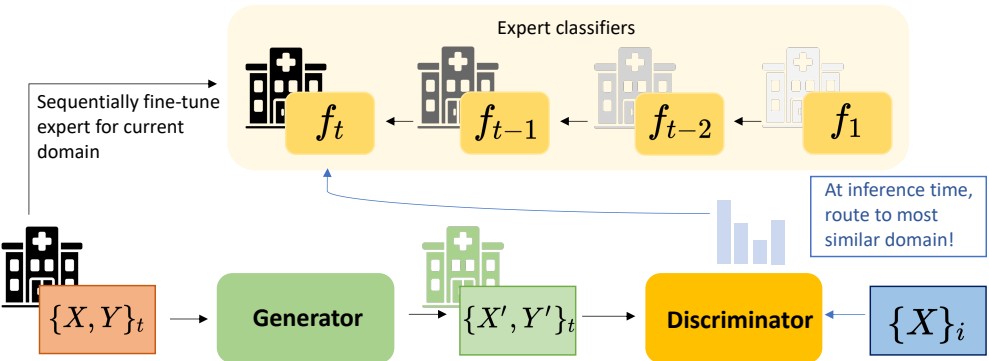

Figure 1: **Generate to Discriminate (G2D)**; During training (black text and arrows), we (i) finetune the generator and expert classifier and (ii) finetune a domain discriminator on synthetic images produced by our generator. At inference time (blue text and arrows), we route test samples to the corresponding expert, based on our discriminator's prediction.

discriminative approaches (Sun et al., 2020), due at least in part to the introduction of noise in the form of low-quality synthetic data.

Without access to a rehearsal buffer, *rehearsal-free* methods often tackle forgetting with dynamically expandable systems with domain-specific parameters, such as expert models (Aljundi et al., 2017; Rypeść et al., 2024) or parameter-efficient prompts (Wang et al., 2022d;c;b). As widely demonstrated in the literature (Wang et al., 2022a;b;d), such modular approaches allow learning parameters independently across domains to potential negative interference, leading to less/no forgetting and better transferring, avoiding tug-of-war scenarios that are exhibited in single, generalist models. However, a core remaining challenge is determining *which* expert or module to invoke at test time. Some recent methods rely on the implicit domain discriminative (zero-shot) capabilities of foundation models (Aharoni & Goldberg, 2020) for an inference-time routing strategy (Wang et al., 2022b). However, as we will show, in settings where the zero-shot discriminative performance of foundation models is limited, these methods underperform (see Table 1). The limitations and above-discussed challenges raise an interesting question: Rather than employing synthetic samples to train the downstream classifier, can synthetic samples serve a more effective purpose in domain discrimination? Can we leverage synthetically generated samples to develop an inference-time routing mechanism?

In this paper, we propose a simple yet effective method for expert routing in domain-incremental learning, *Generate to Discriminate (G2D)*. G2D leverages synthetic data to train a domain discriminator that routes inference-time samples to a set of domain-specific expert classifiers, each checkpointed after each domain is encountered. Surprisingly, we observe that using synthetic data for training a domain router is more effective than using the *same* samples to augment training data for training the label classifier (as in generative replay methods). We also find that our approach outperforms other competitive domain-incremental learning methods, such as replay, regularization-based, and rehearsal-free (prompt-based) methods across the studied benchmarks in both vision and text modalities. To further stress test current methods in more realistic settings, we introduce a new benchmark, DermCL, which consists of a training sequence of four publicly available dermatology medical image classification tasks, each previously utilized independently to evaluate transfer learning or robustness to distribution shifts (Wantlin et al., 2023). We empirically show that competitive prompt-based methods that rely on the implicit (zero-shot) capabilities of foundation models (Aharoni & Goldberg, 2020; Wang et al., 2022b) underperform in these settings, where the downstream task (i.e., skin lesion prediction in minority populations) might diverge from tasks seen in pretraining corpora (see Table 1), emphasizing the need for more realistic benchmarks for a holistic evaluation of continual learning methods.

To summarize, our contributions are as follows:

- We empirically observe that with the *same* synthetic samples, training a domain identifier outperforms augmenting training data for the downstream label classifier—the common approach to leveraging synthetic data in the lifelong learning literature.

- Leveraging this observation, we develop a simple method for expert routing in the domain-incremental learning setting, without *any* access to real data from previously seen tasks, that outperforms considered baselines in both vision and text modalities.

- Towards evaluation of current methods in more realistic settings, we propose a new domain-incremental learning benchmark consisting of a sequence of four real-world, dermatology classification tasks.

## 2 Related Work

The majority of domain-incremental continual learning methods fall into (i) parameter-based and (ii) data-based regularization techniques (Sodhani et al., 2022). We also touch upon more recent approaches, dubbed (iii) prompt-based techniques that leverage pretrained models.

**Parameter-based Regularization Approaches.** Most notable works in this category, namely Elastic Weight Consolidation (EWC; Kirkpatrick et al. (2017)) and Synaptic Intelligence (SI; Zenke et al. (2017)), assess the importance of parameters related to previous domains and use a penalty term to safeguard the knowledge stored in those parameters while updating them for new domains. Another work similarly preserves the knowledge of previous tasks by using the initial task knowledge as a regularizer during training (Li & Hoiem, 2017).

**Data-based Regularization Approaches.** Several competitive methods in the literature retain a subset of data from previous domains as an episodic memory, which is sparsely replayed during the learning of new domains. Each differ in whether the episodic memory is utilized during training, such as GEM (Lopez-Paz & Ranzato, 2017), A-GEM (Chaudhry et al., 2019a), ER (Chaudhry et al., 2019b), MEGA (Guo et al., 2020), or during inference, such as MbPA (de Masson D'Autume et al., 2019; Wang et al., 2020). These methods assume access to retaining true data from previous domains. Yet, this assumption is often violated in practice. Towards operating under more practical assumptions, deep generative replay-based methods have been introduced (DGR; Shin et al. (2017), LAMOL; Sun et al. (2020), LFPT5; Qin & Joty (2022)), where generative models are used to generate synthetic samples for experience replay, often referred to as generative replay. However, these approaches have largely under-performed state-of-the-art discriminative approaches (Sun et al., 2020), due at least in part to the introduction of noise in the form of low-quality synthetic samples.

**Prompt-based and Modular Approaches.** Without access to a rehearsal buffer, *rehearsal-free* methods tackle forgetting with dynamically expandable systems with domain-specific parameters, such as expert models (Aljundi et al., 2017; Rypeść et al., 2024; Li et al., 2024; Park, 2024) or parameter-efficient prompts (Wang et al., 2022b;c; 2024; Le et al., 2024; Yu et al., 2024). [1] The key challenge in such approaches is determining which expert or module to invoke at inference time. Some methods assume access to task identity at test time (De Lange et al., 2021; Li et al., 2019; Wang et al., 2022a) or access to previous samples (Doan et al., 2023; Araujo et al., 2024). However, these assumptions generally restrict practical use in settings where data sharing between institutions are prohibited. In response to the increasing popularity of pretrained models, recent methods rely on the implicit domain discriminative (zero-shot) capabilities of foundation models (Aharoni & Goldberg, 2020; Zhou et al., 2024) for an inference-time routing strategy (Wang et al., 2022b). Mehta et al. (2023) demonstrates that pretrained initializations implicitly mitigate the issue of forgetting when sequentially finetuning models. A more recent line of work, known as prompt-based continual learning, exemplified by L2P (Wang et al., 2022d), DualPrompt (Wang et al., 2022c), S-Prompts (Wang et al., 2022b), and CODA-Prompt (Smith et al., 2023b), involves learning a small number of parameters per domain in the form of continuous token embeddings or prompts while keeping the remaining pretrained model fixed. Although such methods allow continual learning without rehearsal, they depend on access to pretrained models that provide a high-quality backbone across all domains, which may not be available in sensitive environments in real-world deployment (e.g., healthcare). While many methods have been proposed, it is unclear whether they will work well in real-world deployment settings when their implicit assumptions are violated.

---

[1]More broadly, modular approaches utilizing expert models have demonstrated effectiveness across various settings (Zhou et al., 2022), including (but not limited to) multi-task learning (Hazimeh et al., 2021; Fan et al., 2022).

## 3 Preliminaries

In domain-incremental continual learning, the primary goal is to learn a model that adapts to each new domain while mitigating catastrophic forgetting on previously seen domains (Van de Ven & Tolias, 2019). Formally, we consider a sequence of $T$ domains, $\mathcal{D}_1 \rightarrow \cdots \rightarrow \mathcal{D}_T$, where $\mathcal{D}_t = \{x_i^t, y_i^t\}_{i=0}^{N_t}$ represents a dataset corresponding to domain $t$, sampled from an underlying distribution $P_t(\mathcal{X}, \mathcal{Y})$. $x_i^t \in \mathcal{X}$ is the $i$-th image or text passage and $y_i^t \in \mathcal{Y}$ is its label. $N_t$ is the total number of samples for domain $t$. Under this premise, $\forall t$, the marginal or conditional distributions over $\mathcal{X}$ and $\mathcal{Y}$ can change, i.e., $P_t(\mathcal{X}) \neq P_{t+1}(\mathcal{X})$ and $P_t(\mathcal{Y}|\mathcal{X}) \neq P_{t+1}(\mathcal{Y}|\mathcal{X})$, while the label space $\mathcal{Y}$ remains fixed across all domains. The goal is to learn a predictor $f_\theta : \mathcal{X} \rightarrow \mathcal{Y}$, a neural network parameterized by $\theta \in R^P$, to minimize the average expected risk $\hat{R}_T$ across all $T$ domains

$$\hat{R}_T(f_\theta; \mathcal{D}_1, \cdots, \mathcal{D}_T) = \frac{1}{T} \sum_{t=1}^{T} \frac{1}{N_t} \sum_{i=0}^{N_t} \ell(f_\theta(x_i^t), y_i^t), \tag{1}$$

where $\ell$ is a loss function. As standard in the literature, we take $\ell$ as the cross entropy loss.

Since the predictor $f_\theta$ only has access to the current data $\mathcal{D}_t$ during each phase $t$ of the training sequence, this prevents the direct minimization of the average expected risk (Equation 1). If a model sequentially trains by focusing only on minimizing the empirical risk of the current domain, it risks catastrophic forgetting of knowledge acquired from previous domains. Therefore, to demonstrate the model's learning behavior over the sequence of domains and analyze catastrophic forgetting of the previously seen domains, we evaluate the model after training on a specific domain $t$ using the test dataset of that domain, $\mathcal{D}_t^{test} \sim P_t(\mathcal{X}, \mathcal{Y})$, and all test datasets from previously seen domains, $\mathcal{D}_i^{test} \sim P_i(\mathcal{X}, \mathcal{Y}), \forall i \in [1, \ldots, t-1]$. We remark that the domain identity is known only during sequential training and not during inference-time. The goal of domain-incremental continual learning is to produce a system that performs well on test samples randomly drawn from all previously seen domains. Let $\alpha_{s,t}$ denote the accuracy on domain $s$ after training on domain $t$. Following prior work (Lopez-Paz & Ranzato, 2017), we compute the *average accuracy* ($A_t$) metric after training on the domain $t$. This is given by

$$A_t = \frac{1}{t} \sum_{s=1}^{t} \alpha_{s,t}. \tag{2}$$

## 4 Generate to Discriminate (G2D)

Motivated by the limitations of previous works (Shin et al., 2017; Sun et al., 2020; Qin & Joty, 2022), rather than employing synthetic data to augment training data to train the downstream label classifier, we ask whether synthetic data can be used to develop an inference-time routing mechanism. Access to an accurate domain discriminator would answer the question of which expert or module to invoke at inference-time, without any access to real data or ground truth annotations from which domain each data is sourced. Alongside this domain discriminator (router), we learn and maintain a set of domain-specific experts, which we invoke during inference-time according to the router's predictions. We refer to our method of using synthetic data to learn an expert routing mechanism as **Generate to Discriminate (G2D)**.

### 4.1 Generation of Synthetic Samples

At each domain $\mathcal{D}_t$, we finetune a generative model $G$ with samples from the current domain $\{(x_i^t, y_i^t)\}_{i=0}^{N_t}$. For our generator for the vision domain, we finetune an off-the-shelf, text-to-image Stable Diffusion (Rombach et al., 2022) model (see §C.1 for further details). For our generator for the text domain, we use a pretrained T5-Large v1.1 model (Raffel et al., 2020) and optimize via prompt tuning (Lester et al., 2021), which learns continuous input token embeddings (see §B.1 and §C.2 for further details). After finetuning these models on each domain, we then sample synthetic examples $\mathcal{M}_t$ from $G$, which will be used in our approach for learning an expert routing mechanism, as well as in the comparison to generative replay methods. We note that after sampling the synthetic samples from $\mathcal{M}_t$, we discard $G$. In other words, at each domain $D_t$, our method only assumes having access to the current domain's generator $G$ and does not require access to previous ones.

### 4.2 Training Expert Classifiers and the Routing Function

Given synthetically generated data from our generators on domains $\mathcal{D}_1, ..., \mathcal{D}_t$, we finetune a domain discriminator (router), $D_{\theta_t}$ on the union of the synthetically generated samples $\mathcal{M}_1 \cup \cdots \mathcal{M}_{t-1} \cup \mathcal{M}_t$, for domain identity prediction (i.e., $t$-way classification). More formally, we construct a dataset of $\bigcup_{i=1}^{t}\{(x,i)|x \in \mathcal{M}_i\}$ to train a domain discriminator $D_{\theta_t} : \mathcal{X} \rightarrow \mathcal{Y}$, where $\mathcal{Y} = \{1, ..., t-1, t\}$. In short, our domain discriminator learns to predict domain membership, or route samples to their corresponding or most similar domains.

At each domain $\mathcal{D}_t$, we sequentially finetune our classifier $f_{\theta_t}$ as the expert on domain $t$, and add $f_{\theta_t}$ to our list of experts $\{f_{\theta_1}, ...., f_{\theta_{t-1}}, f_{\theta_t}\}$. At inference time, we use our domain discriminator to predict the most likely domain identity $t$, and the test sample is routed to the corresponding expert classifier $f_{\theta_t}$ for our final class prediction (see Figure 1 for a schematic visualization of our approach). For our expert classifiers and domain discriminator for the vision domain, we use a vision transformer (ViT B-16) (Dosovitskiy et al., 2020) pretrained on ImageNet (Deng et al., 2009). For the text domain, we use a pretrained BERT-Base (Kenton & Toutanova, 2019) backbone. To select our hyperparameters, we use the source hold-out performance to select the best combination of parameters (see §C.1 and §C.2 for further details).

---

**Algorithm 1:** Generate to Discriminate (G2D)

**Input:** A stream of domains $\mathcal{D}_1, \ldots, \mathcal{D}_T$, which are only accessed one at a time
 1: **for** $t = 1$ to $T$ **do**
 2:     Get access to current domain $\mathcal{D}_t$
 3:     **(1) Finetune Generator:** Train a domain-specific generator $G_t$ on current domain data $\mathcal{D}_t$
 4:     **(2) Generate Synthetic Data:** Sample synthetic set $\mathcal{M}_t$ from $G_t$
 5:     **(3) Train Expert Classifier:** Finetune expert classifier $f_{\theta_t}$ on real data from $\mathcal{D}_t$
 6:     Add $f_{\theta_t}$ to expert pool $\{f_{\theta_1}, \ldots, f_{\theta_t}\}$
 7:     **(4) Train Domain Discriminator:**
 8:     Construct synthetic dataset $\bigcup_{i=1}^{t}\{(x,i) \mid x \in \mathcal{M}_i\}$
 9:     Train a domain discriminator $D_\theta$ on this dataset to predict domain label $i \in \{1, \ldots, t\}$
10: **end for**
11: **Inference:**
12: Given a test input $x$:
13:     Predict domain $\hat{t} = D_\theta(x)$
14:     Route $x$ to expert $f_{\theta_{\hat{t}}}$ for final label prediction

---

## 5 Experimental Setup

We compare our approach to generative replay approaches and other domain-incremental approaches across a variety of domain-incremental learning benchmarks in both vision and text modalities.

### 5.1 Datasets and Metrics

For our vision experiments, we look at standard domain-incremental benchmarks: DomainNet (Peng et al., 2019) and CORe50 (Lomonaco & Maltoni, 2017), and DermCL (see §5.1), our newly introduced benchmark curated from real-world dermatology tasks (Tschandl et al., 2018; Cassidy et al., 2022; Pacheco et al., 2020; Daneshjou et al., 2022). We use the standard ordering for DomainNet: real, quickdraw, painting, sketch, infograph, and clipart. For CORe50, we use the standard setting of a sequence of 8 domains, with three fixed test domains that are out-of-distribution (OOD). On this task, we remark that learning a domain discriminator essentially learns a similarity function between domains and the OOD test example, routing each new example to the expert trained on the most similar in-distribution (ID) domain. For both datasets, we report average accuracy averaged over 5 random seeds.

For our text experiments, we evaluate our method on the standard domain-incremental question-answering (QA) benchmark as introduced in the work of de Masson D'Autume et al. (2019). The benchmark consists of three QA datasets: SQuAD v1.1(Rajpurkar et al., 2016), TriviaQA (Joshi et al., 2017) and QuAC (Choi et al., 2018). TriviaQA has two sections, Web and Wikipedia, which are considered as separate datasets. We

use four different orderings of domain sequences (see §A.2). Following prior works (de Masson D'Autume et al., 2019; Wang et al., 2020), we compute the $F_1$ score for the QA task and evaluate the model at the end of all domains, i.e., we compute $A_4$.

**DermCL Benchmark.** The lack of realistic benchmarks in the continual learning community and the artificial temporal variation in existing benchmarks have been highlighted in the literature (Lomonaco & Maltoni, 2017; Lin et al., 2021; Wang et al., 2022c). Towards a more holistic evaluation of existing methods on realistic variation in domain shifts, we propose a new domain-incremental continual learning benchmark (**DermCL**), which presents *real world distribution shifts* in dermoscopic images, due to differences in patient demographics, dataset collection period, camera types, and image quality. The benchmark spans four domains of dermoscopic image datasets – HAM10000 (Tschandl et al., 2018), BCN2000 (Cassidy et al., 2022), PAD-UEFS-20 (Pacheco et al., 2020), and DDI (Daneshjou et al., 2022), for a classification task over 5 unified labels of skin lesions. Notably, DDI has been attributed to exhibiting a large performance drop, due to the presence of more dark skin tones from minority groups, exemplifying how failures of backward transfer (i.e., catastrophic forgetting) can have catastrophic outcomes. All four datasets in the sequence are publicly available (see §A.3 for details). We believe that this provides an important evaluation for modern domain-incremental learning approaches, as they often rely on large pretrained models that may have very different pretraining datasets from such real world tasks (e.g., medical imaging).

### 5.2 Baselines

**Generative Replay.** A common way to use synthetic data in lifelong learning is as a buffer of limited synthetic samples used to train the label classifier, introduced by Shin et al. (2017), often referred to as Generative Replay. More precisely, at domain $D_t$, the generator $G_t$ is finetuned with samples from the current domain $\{(x_i^t, y_i^t)\}_{i=0}^{N^t}$, generates synthetic samples $\mathcal{M}_t$ from $G_t$, and stores these samples in a replay buffer. At domain $D_{t+1}$, the classifier $f_\theta$ is sequentially trained on the union of synthetic samples from previous domains $\bigcup_{i=1}^{t-1}\{(x, i)|x \in \mathcal{M}_i\}$ and real samples from the current domain $\{(x_i^t, y_i^t)\}_{i=0}^{N^t}$. We include specific implementation details in §B.2.

**Other Baselines.** First, we assess vanilla sequential finetuning (**SeqFT**), which does not employ any continual learning regularization techniques. Second, we compare with a traditional, parameter-based regularization method, elastic weight consolidation (**EWC**; Kirkpatrick et al., 2017). For vision experiments, we further compare with recent advances in prompt-based methods which have greatly boosted state-of-the-art performance across many vision benchmarks. Specifically, we compare our method with **L2P** (Wang et al., 2022d) and **S-Prompts** (Wang et al., 2022b), as they exhibit competitive performance on domain-incremental learning tasks.[2]

**Oracle and Quasi-Oracle Baselines.** We also compare with the following *oracle* baselines. First, we compare against the setting of when access to real data from *all domains* is allowed at every task step, termed the multi-task learning (**MTL**) baseline. This is equivalent to training on the union of all existing data and can be viewed as an upper bound on performance (i.e., oracle performance) when there is no significant negative transfer between domains. Next, we compare against the following *quasi-oracle* baselines. We compare against a setting we term **Oracle Router**, where we have access to all real examples simultaneously in training the expert router (as opposed to ours in G2D trained via synthetic samples). We compare against **Experience Replay** (**ER**; Chaudhry et al. (2019b)), a data-based regularization method, where the buffer retains limited *real* samples for previously seen domains. For both ER and Generative Replay, we follow the precedent set in the work of de Masson D'Autume et al. (2019), retaining and sampling samples in proportion to the dataset sizes. In the text domain, we additionally compare our approach with methods that leverage replay buffers with real samples for task-specific test-time adaptation, namely Memory-based Parameter Adaptation++ (**MbPA++**) (de Masson D'Autume et al., 2019) and **Meta-MbPA** (Wang et al., 2020). Meta-MbPA trains the model to attain a more suitable initialization for test-time adaptation and achieves the state-of-the-art performance on the Question-Answering benchmark. Note that our setup prevents access to real samples or domain identities from previous domains, making these oracle and quasi-oracle baselines like Experience Replay, MbPA++, Meta-MbPA, Oracle Gate, and MTL inapplicable in practice. However, we

---

[2]We note that there exist two variants of S-Prompts (ViT-based, and CLIP-based). For a fair comparison, we compare with the ViT-based variant that uses the same underlying pretrained checkpoint as the other baselines and G2D.

Table 1: Vision Results. For DomainNet and CORe50, we report performance in terms of average accuracy. For DermCL, performance is reported in terms of average AUC (due to high label imbalance). † denotes results obtained from Wang et al. (2022b). For Experience Replay and Generative Replay, we use a fixed buffer size of 15/class for DomainNet and 100/class for DermCL, according to the lower bound on samples per class in the actual dataset; and 50/class for CORe50, following previous practice (Wang et al., 2020).

| Method | DomainNet | CORe50 | DermCL |
|---|---|---|---|
| Seq-FT | $57.45 \pm 0.36$ | $83.10 \pm 0.50$ | $70.49 \pm 2.51$ |
| EWC | $57.25 \pm 0.18$ | $82.10 \pm 1.89$ | $75.26 \pm 0.69$ |
| L2P | $40.2^{\dagger}$ | $78.3 \pm 0.1^{\dagger}$ | $74.20 \pm 1.42$ |
| S-Prompts | $56.13 \pm 0.39$ | $83.38 \pm 0.36$ | $81.59 \pm 0.22$ |
| Generative Replay | $60.16 \pm 0.64$ | $92.50 \pm 0.40$ | $81.23 \pm 1.12$ |
| **G2D (Ours)** | $\mathbf{70.03 \pm 0.02}$ | $\mathbf{94.05 \pm 0.53}$ | $\mathbf{85.24 \pm 0.85}$ |
| Oracle Router | $71.24 \pm 0.18$ | $95.44 \pm 0.11$ | $91.92 \pm 0.31$ |
| Experience Replay | $65.64 \pm 0.47$ | $93.65 \pm 0.60$ | $89.17 \pm 0.79$ |
| Upper Bound (MTL) | $75.10 \pm 0.09$ | $97.19 \pm 0.12$ | $93.34 \pm 0.96$ |

still include these baselines for a more comprehensive comparison to demonstrate how G2D performs against methods that (i) retain access to *real* samples from previous domains or (ii) assume access to ground-truth domain identity.

## 6 Results

We observe that G2D outperforms competitive methods in the considered vision and text benchmarks: DomainNet, CORe50, DermCL, and Question Answering (Table 1, 2). Notably, on datasets characterized by significant domain shifts that amplify negative backward transfer, such as DomainNet and DermCL,[3] our method achieves substantial performance improvements. Additionally, G2D demonstrates strong out-of-distribution (OOD) performance compared to prior baselines, as evidenced by results on CORe50, where evaluation encompasses three OOD datasets (e.g., domains never seen during training). Amongst the con-

Table 2: Text Results. Performance is reported in terms of average $F_1$ (averaged over 4 domain sequences). ‡ denotes results obtained from Wang et al. (2020). ER, MbPA++ and Meta-MbPA uses a buffer size of 1% actual samples.

| Method | Question Answering |
|---|---|
| Seq-FT | $56.6 \pm 5.7$ |
| EWC | $55.9 \pm 3.7$ |
| Generative Replay | $59.5 \pm 0.9$ |
| **G2D (Ours)** | $\mathbf{64.7 \pm 0.2}$ |
| Oracle Router | $65.4 \pm 0.0$ |
| Experience Replay | $62.6 \pm 1.4$ |
| MbPA++ | $61.9 \pm 0.2^{\ddagger}$ |
| Meta-MbPA | $64.9 \pm 0.3^{\ddagger}$ |
| Upper Bound (MTL) | $68.6 \pm 0.0$ |

sidered baselines, the comparison with Generative Replay is particularly instructive for understanding how to most effectively utilize synthetic data in the domain-incremental learning setting. In this comparison, we use the same set of synthetic samples, but in different ways.[4] Our results demonstrate that using the same set of synthetic samples[5] for domain discrimination consistently outperforms their use for augmenting training data for downstream classification (i.e., Generative Replay) across all considered benchmarks (see Table 1 and Table 2). Fundamentally, G2D relies on the generator to accurately model samples from different domains in order to train the domain discriminator. Generative Replay, on the other hand, requires an accurate modeling of downstream class labels rather than domains. Thus, our results empirically suggest that modeling the difference in domains is a relatively easier task than modeling the difference in downstream classes (for the considered generative models).

---

[3]DomainNet contains highly heterogenous domains, as highlighted by prior works (Wang et al., 2022b). DermCL exhibits substantial domain shifts, due to its inclusion of patient populations with diverse racial backgrounds and skin tones.

[4]We include the specific details and implementation of Generative Replay in §B.2.

[5]In Table 12 and Figure 3 (see Appendix §F), we include example visualizations of generated samples for both modalities.

Table 3: We report the accuracy of domain discrimination (expert routing) for the considered baselines to complement the analysis presented in Figure 2 and §6.1. Note the oracle discriminator refers to a model trained on real samples, serving as the reference upper bound. "-" denotes not applicable.

| Dataset | S-Prompts Disc. | G2D Disc. (ours) | Oracle Disc. |
|---------|-----------------|-------------------|--------------|
| CORe50  | $90.81 \pm 0.27$ | $97.38 \pm 0.93$ | $98.94 \pm 0.07$ |
| QA      | -                | $94.5 \pm 0.2$   | $97.1 \pm 0.0$ |

Figure 2: (Three left-most plots) Domain discrimination visualizations for CORe50 benchmark (8 domains). From left to right, the first subplot shows how domains are clustered via S-Prompts (Wang et al., 2022b). In the second and third subplots, the clusterings for our domain discriminator and the oracle discriminator are visualized. (Three right-most plots) Visualizations of domain clusterings for the QA benchmark (4 domains). The first subplot highlights the implicit domain discriminative nature of pretrained BERT-Base representations (Kenton & Toutanova, 2019). In the second and third subplots, we visualize the clustering of representations from our discriminator and the oracle discriminator trained using real samples. Interestingly, the discriminators trained with synthetic samples closely mirrors the performance and clustering patterns of the discriminators trained using real data.

While our motivation to operate under more practical assumptions restricts data sharing across domains, we also include comparisons with competitive methods that assume access to real data from previously seen domains, for a more comprehensive evaluation. This allows us to understand the utility of our method even when such data constraints are relaxed. We observe that although our approach operates without access to real data from previously seen domains, it retains competitive performance with Experience Replay (see Table 1 and Table 2), and even outperforms the previous state-of-the-art in test-time adaptation-based continual learning, Meta-MbPA on text experiments (see Table 2). These findings underscore the ability of our method to enhance performance even in scenarios not characterized by stringent constraints on data sharing.

## 6.1 Analysis of Expert Routing Performance

To better understand our observed empirical gains, we examine our domain discriminator (expert router) independently from the overall pipeline. For this purpose, we make a departure from the downstream classification task and focus solely on analysis of the expert routing procedure.

For the vision domain, we assess the performance of our discriminator with the (1) oracle discriminator (trained on a buffer of real samples) and (2) S-Prompts (Wang et al., 2022b), a previous state-of-the-art domain-incremental learning method that leverages a routing mechanism. S-Prompts leverages K-Means during training to store centroids for each domain; then during inference time, it employs KNN to identify the domain of a given test image feature by determining the domains of the $K$ nearest centroids. In Figure 2, we present t-SNE plots (Van der Maaten & Hinton, 2008) illustrating domain clusterings between these three methods for CORe50, the benchmark comprised of the longest domain sequence (eight domains). We observe that our method achieves more accurate clustering and domain identification than the S-Prompts method (see Table 3).

For the text domain, we present t-SNE plots visualizing the domain discriminative capability (Aharoni & Goldberg, 2020) of a pretrained language model and the G2D discriminator trained on synthetic samples (see

Figure 2). Notably, in the pretrained language model's clustering, there is confusion between the TrWeb (orange) and TrWiki (green) domains, both derived from the same TriviaQA dataset (Joshi et al., 2017). Similarly, the TrWiki (green) and SQuAD (red) domains, originating from the same Wikipedia source, necessitate explicit discriminator training. We clearly observe that training an explicit discriminator results in more accurate clustering. This is remedied in the clustering of the G2D discriminator, closely matching the behavior of the upper bound, namely a discriminator trained using real samples (see Table 3). Overall, for both vision and text domains, we observe that G2D improves domain identifiability (expert routing) and demonstrates competitive performance to a discriminator trained on *real* data, exhibiting similar clustering patterns.

## 7 Discussion, Limitations, and Conclusion

In this work, we propose a simple yet effective method for domain-incremental learning that leverages synthetic data to train a domain discriminator, enabling an inference-time routing mechanism. We establish that this approach is more effective than using the same synthetic samples to directly train the downstream label classifier (as past works have explored). While we do not claim that this phenomenon is universal, our findings consistently held true across many different domain-incremental learning benchmarks we investigated for both modalities. We leave a deeper theoretical investigation behind the mechanisms of this empirical finding for future work. Further, our analysis of the capabilities of our domain discriminator, which encompasses existing domain discrimination approaches, unsupervised clustering methods, and a discriminator trained on real samples, finds that our method improves expert routing performance across both modalities. Building on this, expanded formulations of experts and modular architectures, such as mixture of experts (Shazeer et al., 2017) and its derivatives thereof (e.g., multi-gate mixture of experts (Ma et al., 2018)), offer intriguing possibilities for complementary extensions of our method, which we leave for future study. Motivated by our interest in real-world settings (e.g., clinical healthcare), where the prohibition on data sharing is especially well-motivated, and by the lack of realistic benchmarks in the continual learning community more broadly,[6] we introduce a new domain-incremental learning benchmark to further stress-test competing methods. In addition to its practical relevance in the problem setting at hand, our observations provide interesting insights into different perspectives on the use of synthetic data in such scenarios where real data cannot be shared.

**Limitations.** A potential limitation of our method is that the number of experts scales linearly with the number of domains encountered. While this introduces potential computational overhead, it is unlikely to be prohibitive in our motivating scenarios (i.e., healthcare), where the number of domains (e.g., institutions) is typically modest, often on the order of tens. This ensures the method remains practical for real-world applications, though strategies to address scaling for larger domain counts warrant further exploration. We report a detailed breakdown of the computational cost and storage requirements of our method in Appendix B.3. Further, we remark that there exist potential privacy concerns, when leveraging generative models to sample synthetic data, where memorization could potentially reveal sensitive information from the original data in the training dataset. For these concerns, there is an important, separate field of work in improving the differential privacy (Dwork & Roth, 2014; Dwork et al., 2006) of such methods, by training these models with privacy constraints (Dockhorn et al., 2022; Lyu et al., 2023; Cao et al., 2021). In this work, we do not take a stand on when or whether the sharing of generative models (or synthetic samples) should be permissible. While this is an important issue to be considered prior to real-world deployment of any method that involves generative models, how institutional practices develop and how the regulatory environment evolves will be informed, to a large degree, by exploratory research that characterizes both (i) the potential benefits; and (ii) the potential risks associated with the dissemination of generative models trained on real data. We see our research as helping to elucidate the potential benefits of synthetic data in such settings.

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

# A  Dataset Details

In this section, we provide additional details about the datasets used in our experiments. We provide the overall dataset statistics in Table 4.

## A.1  Overall Dataset Statistics

Table 4: Per-domain statistics for the domain-incremental learning benchmarks used in both vision and text experiments.

| Dataset | Domain | Train | Validation | Test |
|---|---|---|---|---|
| **CORe50** Lomonaco & Maltoni (2017) | 0 | 13491 | 1499 | 44972 |
| | 1 | 13495 | 1500 | 44972 |
| | 2 | 13487 | 1499 | 44972 |
| | 3 | 13494 | 1499 | 44972 |
| | 4 | 13490 | 1499 | 44972 |
| | 5 | 13490 | 1499 | 44972 |
| | 6 | 13469 | 1497 | 44972 |
| | 7 | 13485 | 1498 | 44972 |
| **DomainNet** Peng et al. (2019) | Real | 108814 | 12091 | 52040 |
| | Quickdraw | 108674 | 12075 | 51749 |
| | Painting | 45373 | 5041 | 21849 |
| | Sketch | 43389 | 4821 | 20915 |
| | Infograph | 32419 | 3602 | 15581 |
| | Clipart | 30171 | 3352 | 14603 |
| **DermCL** | HAM10000 Tschandl et al. (2018) | 7210 | 802 | 2003 |
| | BCN2000 Cassidy et al. (2022) | 18238 | 2027 | 5066 |
| | PAD-UEFS-20 Pacheco et al. (2020) | 1655 | 184 | 459 |
| | DDI Daneshjou et al. (2022) | 419 | 105 | 132 |
| **Question Answering** de Masson D'Autume et al. (2019) | SQuAD 1.1 Rajpurkar et al. (2016) | 81000 | 9000 | 10000 |
| | TriviaQA (Web) Joshi et al. (2017) | 68400 | 7600 | 10000 |
| | TriviaQA (Wiki) Joshi et al. (2017) | 54000 | 6000 | 8000 |
| | QuAC Choi et al. (2018) | 72000 | 8000 | 7000 |

## A.2  Question Answering Task Orders

For our Question-Answering (QA) benchmark de Masson D'Autume et al. (2019), we consider the following four sequences of dataset orderings for a more comprehensive evaluation. We report the average accuracy over all sequences.

1. QuAC → TriviaQA (Web) → TriviaQA (Wiki) → SQuAD

2. SQuAD→ TriviaQA (Wiki) → QuAC → TriviaQA (Web)

3. TriviaQA (Web) → TriviaQA (Wiki) → SQuAD → QuAC

4. TriviaQA (Wiki) → QuAC → TriviaQA (Web) → SQuAD

## A.3  DermCL Benchmark Details

This benchmark offers a sequence of four dermatology imaging tasks. Distribution shifts are present across all four domains (HAM10000 Tschandl et al. (2018), BCN2000 Cassidy et al. (2022), PAD-UEFS-20 Pacheco et al. (2020), and, DDI Daneshjou et al. (2022)), in both demographics and data collection techniques. Dermoscopic image classification on diverse patient populations present complexities arising from (but not limited to) intraclass variations encompassing lesion texture, scale, and color. The label space for DermCL is defined as the following 5 unified labels: MEL (melanoma), NEV (nevus), BCC (basal cell carcinoma), AKIEC (includes actinic keratoses, intraepithelial carcinoma, squamous cell carcinoma), and Other diseases. We provide additional details about each domain in the following Table 5. All four datasets in the sequence are publicly available at `https://github.com/rajpurkarlab/BenchMD` Wantlin et al. (2023) (for HAM10000, BCN2000, PAD-UEFS-20) and `https://ddi-dataset.github.io` Daneshjou et al. (2022) (for DDI).

Table 5: Per-domain information for the DermCL Benchmark.

| Domain | Description |
| --- | --- |
| **HAM10000**Tschandl et al. (2018) | The HAM10000 dataset was collected over the past 20 years from hospitals in Austria and Australia using dermatoscopes. |
| **BCN2000**Cassidy et al. (2022) | The BCN2000 dataset was collected from Spanish hospitals between 2010 and 2016 using dermatoscopes. |
| **PAD-UEFS-20**Pacheco et al. (2020) | The PAD-UEFS-20 dataset was collected from Brazilian hospitals in 2020 using smartphone cameras. |
| **DDI**Daneshjou et al. (2022) | The DDI dataset contains images collected from pathology reports in Stanford Clinics from 2010-2020, representing 570 unique patients with diverse skin tone representation. It is notable for inducing significant performance drops in models, due to the presence of more dark skin tones and uncommon diseases. |

# B  Additional Implementation Details

In this section, we provide additional details about the (1) implementation of our method, (2) implementation of generative replay, (3) computational costs, and (4) hardware requirements.

## B.1  G2D Implementation Details

**Vision.**  For our generator, we use an off-the-shelf, text-to-image Stable Diffusion (Rombach et al., 2022) for our generative model (see Appendix §C.1 for further hyperparameter details). During inference, we prompt the text-to-image conditional diffusion model with the text label to sample generations. For our expert classifiers and domain discriminator, we use a Vision Transformer (ViT B-16) (Dosovitskiy et al., 2020), initialized with the pretrained ImageNet (Deng et al., 2009) checkpoint. For our hyperparameter search, we use the source hold-out performance to select the best combination of parameters (see Appendix §C.1 for further details). For parameter efficiency, we finetune our expert classifiers with low-rank adaptation (i.e., LoRA) (Hu et al., 2021), where we only adapt the attention weights and keep the remaining parameters of the UNet architecture frozen.

**Text.**  For our generator, we use prompt tuning to learn the parameter-efficient models (Lester et al., 2021). We use the pretrained T5-Large v1.1 checkpoint adapted for prompt tuning as the backbone (Raffel et al., 2020) and the prompt embeddings are initialized randomly. We input a special token into the model and conditionally generate a document content, question, and answer, all separated by the special tokens. During the generation process, we provide multiple text prompts. We use the following text prompts to conditionally generate synthetic samples:

```
text_prompts = [
    "Generate article, question and answer.",
    "Generate context, question and answer.",
    "Generate answers by copying from the generated article.",
    "Generate factual questions from the generated article."
]
```

During generation, we use ancestral sampling, which selects the next token randomly based on the model's probability distribution over the entire vocabulary, thereby reducing the risk of repetition. We generate samples with a minimum length of 50 tokens and a maximum of 1,000 tokens and retain only those samples that contain exactly one question-answer pair with the answer included in the generated document content. For training our domain discriminator and expert classifiers, we use low-rank adaptation (LoRA; Hu et al., 2021) and freeze the pretrained BERT-Base (Kenton & Toutanova, 2019) backbone (see Appendix §C.2 for further details).

### B.2 Generative Replay Implementation Details

We implement Generative Replay following standard practice (Shin et al., 2017) with the following updates to address limitations of the original approach: (1) The generative model originally used by Shin et al. (2017), WGAN-GP, is quite outdated. To ensure a fair comparison, we revisit Generative Replay using the same generative models employed in our method (G2D) — Stable Diffusion (Rombach et al., 2022) for vision data and T5 (Raffel et al., 2020) for text data. (2) In the original Generative Replay implementation, the generator is sequentially finetuned (in addition to the classifier) on each domain in the sequence. This introduces the challenge of catastrophic forgetting in the generative model and introduces additional artifacts caused by continually training on noisy samples. In order to address this issue, there is an important line of work Smith et al. (2023a) focused on investigating catastrophic forgetting *within the learning procedure of generative models*, but such investigations fall beyond the scope of this study. Instead, for the purposes of this work, we mitigate this issue by finetuning the generator from the pretrained checkpoint rather than the finetuned checkpoint from the previous domain. This is applied consistently to both G2D and Generative Replay to ensure a fair comparison. In summary, these adjustments allow us to compare the performance of Generative Replay with our method, ensuring both approaches utilize the **same fixed set of synthetic samples**.

### B.3 Computational Cost and Storage Requirements

We acknowledge that our method incurs additional computational and storage costs, and practitioners should weigh these factors when choosing between approaches. Compared to naive sequential fine-tuning (Seq-FT), our method adds a total of 5-8 hours (per domain) of compute time on a single A6000 GPU, due to fine-tuning and sampling from the generator, and 1-3 hours (per domain) for training the domain discriminator. However, we note that this can all be done in parallel with expert training (that takes 1-8 hours on each domain as is done in Seq-FT) and, thus, our approach incurs additional total time of up to 4-7 hours per domain. In terms of storage, the main overheads come from (1) storing synthetic samples (e.g., 2.4GB for a buffer of 5,000 samples), and (2) retaining fine-tuned expert classifiers ($\sim$330MB each). For the generator and discriminator, we only retain the current domain's models—$\sim$18GB and $\sim$330MB respectively—rather than storing all previous versions. Overall, we believe the additional resource demands are manageable and not unreasonable relative to practices in recent machine learning workflows. A detailed breakdown of these requirements is provided in Table 6.

Table 6: Computational and storage requirements for each component. Seq-FT denotes standard sequential fine-tuning. For Synthetic Data, (1) the train time denotes sampling time and (2) we report storage in terms of a total buffer size of 5000 samples. We note that the training time differs based on each dataset (and the number of samples in their respective train sets).

|  | Train Resource | Train Time (hrs) | Storage (GB) |
| --- | --- | --- | --- |
| Expert Classifier | 1 A6000 GPU | $1 \sim 8$ / domain | $\sim$330 MB / domain |
| Domain Discriminator | 1 A6000 GPU | $1 \sim 3$ / domain | $\sim$330 MB |
| Generator | 1 A6000 GPU | $4 \sim 6$ / domain | 18 GB |
| Synthetic Data | 1 A6000 GPU | $1 \sim 3$ / domain | $\sim$ 2.4 GB (for 5000 samples) |
| Seq-FT | 1 A6000 GPU | $1 \sim 8$ / domain | $\sim$330 MB |

### B.4 Hardware Requirements

All experiments were conducted using NVIDIA RTX A6000 and NVIDIA RTX 6000Ada graphics cards.

## C Hyperparameter Details

### C.1 Vision Experiment Hyperparameter Details

For all of our baselines, experts, and domain discriminators, we use the same pretrained model checkpoints, namely, the ViT-B/16 checkpoint trained on ImageNet (`vit_base_patch16_224`), released by Pytorch Image

Models (timm). For our generative model, we use Stable Diffusion (Rombach et al., 2022) as our conditional diffusion model with weights from the CompVis/stable-diffusion-v1-4 checkpoint. Hyperparameters are detailed in Table 7 and Table 8. For all hyperparameter searches, we use the source hold-out performance to select the best combination of parameters.

Table 7: Finetuning hyperparameters for the generative models in our vision experiments.

| Hyperparameter | Value |
|---|---|
| Number of steps | 150,000 |
| Resolution | 512 |
| Learning Rate | $1e^{-5}$ |
| Learning Rate Scheduuler | Constant |
| Optimizer | AdamW (Loshchilov & Hutter, 2017) |
| Batch Size | 16 |
| Weight Decay | $1e^{-2}$ |

Table 8: Finetuning hyperparameters for the classifiers in our vision experiments

| Hyperparameter | DomainNet | CORe50 | DermCL |
|---|---|---|---|
| Number of Epochs | $20 \sim 50$ | 10 | 10 |
| Learning Rate | $0.005 \sim 0.01$ | 0.07 | 0.005 |
| Optimizer | SGD | SGD | SGD |
| Momentum | 0 | 0 | 0.9 |
| Batch Size | 128 | 128 | 128 |
| LoRA Rank | 16 | 16 | 16 |
| LoRA Alpha | 16 | 16 | 16 |

To ensure that we are evaluating our baselines comprehensively, we also run a hyperparameter search for different regularization values $\lambda \in [0.5, 1, 10, 100]$ for the Elastic Weight Consolidation (EWC) method, and use $\lambda = 1$. For Experience Replay and Generative Replay baselines, we retain (or sample) 15/class samples for DomainNet and 100/class samples for DermCL, according to the lower bound on samples per class in the actual dataset; and 50/class samples for CORe50, following previous practice (Wang et al., 2020). For all of our baselines, we use the same pretrained model checkpoints, namely, the ViT-B/16 checkpoint trained on ImageNet (`vit_base_patch16_224`), released by Pytorch Image Models (timm).

### C.2 Text Experiment Hyperparameter Details

For our expert classifiers and domain discriminator, we use LoRA and freeze the pretrained BERT-Base (Kenton & Toutanova, 2019) backbone. The BERT-base architecture has 12 Transformer layers, 12 self-attention heads, and 768 hidden dimensions (110M parameters). For our generative model, we use prompt tuning to learn parameter-efficient models (Lester et al., 2021). We use the pretrained T5-Large v1.1 checkpoint adapted for prompt tuning as the backbone (Raffel et al., 2020), and the prompt embeddings are initialized randomly. Hyperparameters are detailed in Table 9 and Table 10. The hyperparameters for baseline methods are set as described in Wang et al. (2020). For Experience Replay (and Generative Replay), we retain (or sample) 1% of examples which account for around 6,000 examples across all four considered domains.

## D Additional Experiments

### D.1 Additional Discriminator Comparisons

We present the results on the remaining datasets for domain discrimination, comparisons between the S-Prompts discriminator, our G2D discriminator, and an oracle discriminator trained on samples with ground

—

Table 9: Finetuning hyperparameters for the generative models in our text experiments. We set the prompt length to 400 tokens as our prompt length, which accounts for 819k trainable parameters (roughly 0.1% of the total number of parameters in T5-Large). For our learning rate scheduler, we use warmup ratio of 0.01 and linear decay over 5 epochs.

| Hyperparameter | Value |
|---|---|
| Prompt Length (Tokens) | 400 |
| Optimizer | Adam (Kingma & Ba, 2014) |
| Learning Rate | 1.0 |
| Batch Size | 8 |
| Weight Decay | $1e^{-5}$ |
| Maximum Sequence Length | 512 |

Table 10: Finetuning hyperparameters for the classifiers in our text experiments.

| Hyperparameter | Question Answering |
|---|---|
| Number of epochs | 5 (discriminator), 3 (experts) |
| Learning rate | $5e^{-4}$ |
| Optimizer | Adam |
| Dropout | 0.1 |
| Batch Size | 8 |
| Max Input Length | 384 |
| LoRA Rank | 32 |
| LoRA Alpha | 32 |

truth domain identifiers. We observe that our discriminator outperforms the alternative approach on all tasks.

Table 11: We compare the performance of the G2D domain discriminator with the S-Prompts discriminator (that uses KMeans and KNNs). We also provide a comparison to a discriminator trained on real samples (Oracle Discriminator). The G2D discriminator outperforms the S-Prompts discriminator on all tasks.

| Dataset | S-Prompts Disc. | G2D Disc. (ours) | Oracle Disc. |
|---|---|---|---|
| DomainNet | $80.33 \pm 0.05$ | $83.74 \pm 0.99$ | $85.03 \pm 0.74$ |
| CORe50 | $90.81 \pm 0.27$ | $97.38 \pm 0.93$ | $98.94 \pm 0.07$ |
| DermCL | $71.22 \pm 0.10$ | $75.22 \pm 0.48$ | $80.45 \pm 3.18$ |

# E    Discussion on Generative Models in Healthcare

In general, and for good reason, practice in healthcare moves considerably slower than exploratory machine learning research. It is generally the case that ideas take root in the research community long before they show up in the clinic. Following this convention, due to the recency of successes of generative models (relative to discriminative models), contractual or regulatory requirements surrounding generative models is still in nascent stages of development.

*What is the current status quo?* The setting where model weights may be shared but not the actual training data is a well-known setting in the healthcare domain (Kamran et al., 2022; Ulloa-Cerna et al., 2022; Walsh et al., 2023). We elaborate on two examples: (1) Kamran et al. (2022) presents a multisite external validation study for early identification of COVID-19 patients at risk of clinical deterioration, which require sharing the model trained on private EHR data from one US hospital with 12 other US medical centers; (2) Ulloa-Cerna et al. (2022) presents a multisite external validation study for model development for identifying patients at increased risk of undiagnosed structural heart disease, which requires sharing the model trained on private

EHR data and patient echocardiography reports from one site with 10 other independent sites. While these are generally examples of discriminative models being shared across facilities as opposed to generative models, this demonstrates the general principle that in such domains, model sharing is often permissible in settings where data sharing is not.

On one hand, it seems intuitive that healthcare institutions might be queasier about sharing generative models than sharing discriminative models. On the other hand,

- Healthcare institutions are even queasier about sharing real data — and to this end there is a large mainstream line of work investigating the use of generative models for direct sharing or for producing synthetic datasets that could be disseminated in lieu of actual patient data (Chen et al., 2021; Coyner et al., 2022; DuMont Schütte et al., 2021)

- From a standpoint of most contractual or regulatory requirements, it is not yet clear even if generative models sit in a different category than discriminative models or if they should follow the same current regulatory requirements for discriminative models.

- How institutional practices develop and the regulatory environment evolve will be informed, to a large degree, by exploratory research that characterizes both (i) the potential benefits and (ii) the potential risks associated with the dissemination of generative models trained on medical data.

## F  Example Generations

We include example generations for both image and text domains.

### F.1  Examples of Generated Images

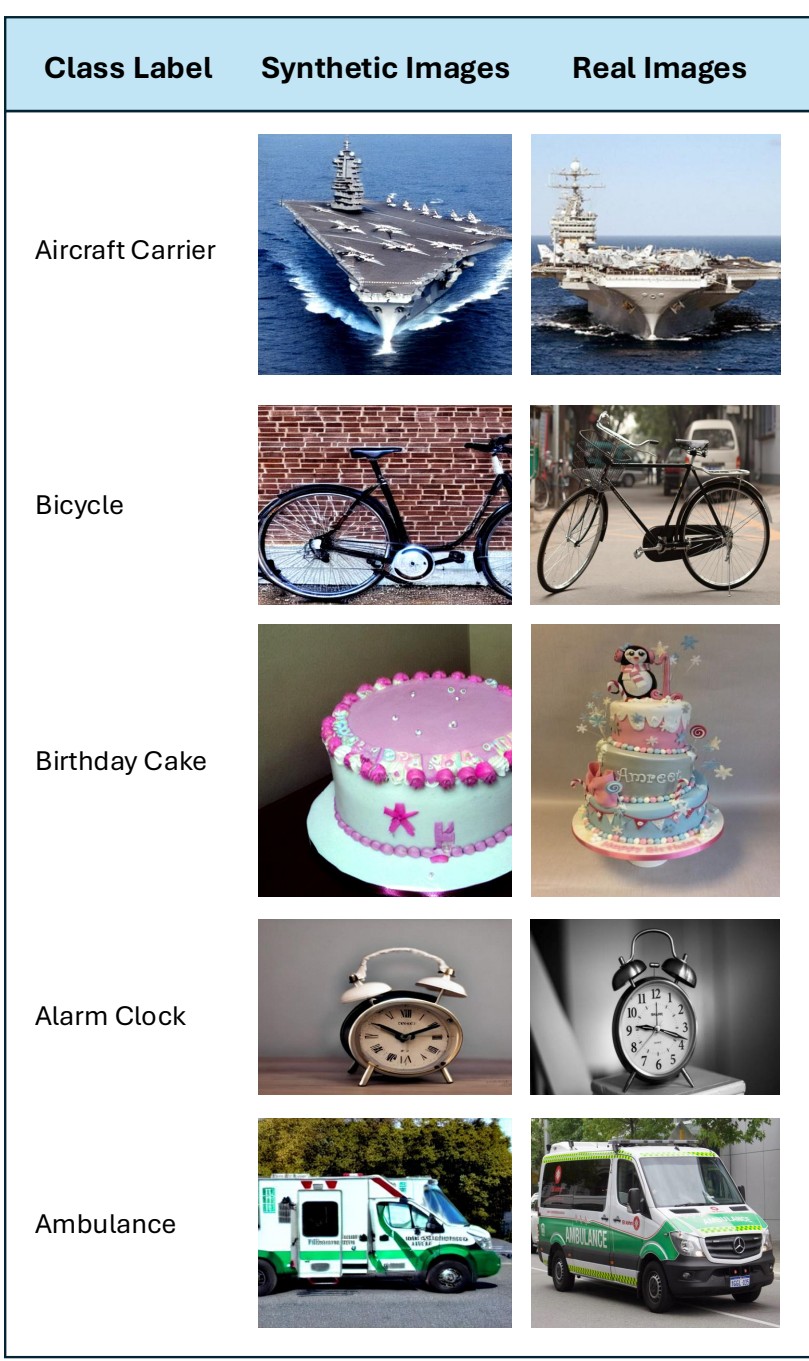

Figure 3: Comparison of sampled synthetic images and real images from the DomainNet benchmark.

## F.2  Examples of Generated Text

Table 12: Generated samples (context, question-answer pair) for the SQuAD domain. For the incorrectly generated samples, we underline one possible correct answer.

| Dataset | Fields | Generated sample |
|---|---|---|
| SQuAD | | **Correct** |
| | *Context:* | During the late 19th and early 20th centuries, the city's trade sector expanded greatly, and through the 20th century, more than half of its residents worked toward higher status in the military. Other industries included industry, commerce, public administration, and medicine. Its largest sector was public services - police, fire services, and healthcare and was the nation's third largest. |
| | *Question:* | What is one of the industry sectors that were the biggest? |
| | *Answer:* | public services |
| | | **Incorrect** |
| | *Context:* | In the United Kingdom there is a general agreement between the government and the private sector in principle that both private and publicly funded institutions of higher education constitute university colleges. Further, there is a mutual agreement between the independent college and the university to promote higher education. However, in both cases all the institutions of higher education are either controlled by private individuals or by a national agency, in such a way as to protect freedom of expression. |
| | *Question:* | What are some of the institutions of higher education that are controlled by private individuals? |
| | *Answer:* | private individuals |

