# OpenReview forum: "Expert Routing with Synthetic Data for Domain Incremental Learning"
_TMLR — Accepted by TMLR_

### Review · Reviewer_RphK · 2025-04-04

**Summary Of Contributions:**

The paper proposes Generate to Discriminate (G2D), a method for domain-incremental learning that uses synthetic data not to augment classifiers directly, but to train a domain discriminator that routes test samples to the appropriate expert model. This routing mechanism allows for effective handling of catastrophic forgetting without access to real data from past domains, outperforming traditional generative replay and regularization-based methods in both vision and language tasks. Additionally, the authors introduce a new benchmark, DermCL, highlighting challenges in medical imaging with domain shifts, and show G2D's strong performance in such realistic scenarios.

**Audience:**

Yes

**Broader Impact Concerns:**

The broader impact of this submission lies in its potential to significantly improve machine learning deployment in real-world, privacy-sensitive domains like healthcare, finance, and autonomous systems, where sharing raw data is often infeasible. By leveraging synthetic data for expert routing instead of classification, the proposed method (G2D) enables effective domain adaptation and lifelong learning without violating data-sharing constraints, making it highly applicable in regulated environments.

**Claims And Evidence:**

Yes

**Requested Changes:**

Please make corresponding modifications based on the weaknesses I mentioned in the review.

**Strengths And Weaknesses:**

Strengthens:
1. The paper introduces a new and effective use of synthetic data—training a domain discriminator for expert routing—rather than the conventional use for data augmentation, which shows consistently better performance.

2. The method operates without access to real data from previous domains, making it suitable for privacy-sensitive applications where data sharing is restricted.

3. The paper includes detailed analysis of domain discrimination performance, showing that synthetic-data-trained discriminators can approach the accuracy of those trained on real data.


Weaknesses:

1. The authors are encouraged to add another subsection in the related work to discuss what are the other research directions conducted on different domains, e.g., unsupervised domain adaptation (e.g., [1], [2]), domain generalization (e.g., [3], [4]), to justify what are the major differences among these different task settings when compared with domain incremental learning. Some other prompts (e.g., [1], [2], [6]) based technique to deal with domain gap needs also be mentioned in the related work.

[1] Bai, S., Zhang, M., Zhou, W., Huang, S., Luan, Z., Wang, D., & Chen, B. (2024, March). Prompt-based distribution alignment for unsupervised domain adaptation. In Proceedings of the AAAI conference on artificial intelligence (Vol. 38, No. 2, pp. 729-737).

[2] Du, Z., Li, X., Li, F., Lu, K., Zhu, L., & Li, J. (2024). Domain-agnostic mutual prompting for unsupervised domain adaptation. In Proceedings of the IEEE/CVF Conference on Computer Vision and Pattern Recognition (pp. 23375-23384).

[3] Peng, K., Wen, D., Yang, K., Luo, A., Chen, Y., Fu, J., ... & Stiefelhagen, R. (2024). Advancing open-set domain generalization using evidential bi-level hardest domain scheduler. Advances in Neural Information Processing Systems, 37, 85412-85440.

[4] Zheng, G., Huai, M., & Zhang, A. (2024, March). AdvST: Revisiting data augmentations for single domain generalization. In Proceedings of the AAAI conference on artificial intelligence (Vol. 38, No. 19, pp. 21832-21840).

[5] Peng, K., Wen, D., Saquib, S. M., Chen, Y., Zheng, J., Schneider, D., ... & Stiefelhagen, R. (2024). Mitigating Label Noise using Prompt-Based Hyperbolic Meta-Learning in Open-Set Domain Generalization. arXiv preprint arXiv:2412.18342.


2. The experiments are only conducted using ViT B, the cross backbone generalization capability needs to be checked. The authors are encouraged to replace the backbone with some CNN architectures to see its performance (e.g., ResNet50).


3.  Per-domain performances are not provided, which are also interesting.


4. Section 4.2 needs more details to make the method clearer. The authors are encouraged to describe it via algorithm format.

5. The number of expert models grows linearly with the number of domains, potentially introducing significant computational and memory overhead in large-scale settings.

6. The study primarily targets classification problems; the applicability of G2D to other task types (e.g., regression, structured prediction) remains unexplored.

---

> ### Author Response · Authors · 2025-05-21
> **Response to Reviewer RphK**
>
> Dear Reviewer RphK,
>
> Thank you for the feedback! We’re grateful for your acknowledgement of the value of our work. We address your comments below.
>
> ### **Additional Related Work**
>
> > The authors are encouraged to add another subsection in the related work to discuss what are the other research directions conducted on different domains, e.g., unsupervised domain adaptation (e.g., [1], [2]), domain generalization (e.g., [3], [4]), to justify what are the major differences among these different task settings when compared with domain incremental learning. Some other prompts (e.g., [1], [2], [6]) based technique to deal with domain gap needs also be mentioned in the related work.
>
> Thanks for the feedback! We have added another subsection in our Related Work section (Section 2) to include a discussion of the key distinctions between the domain incremental learning setting and the noted other research areas—unsupervised domain adaptation and domain generalization. We have also incorporated all the referenced work, including the mentioned prompt-based techniques for handling domain gaps.
>
> ### **Additional experiments**
> > The experiments are only conducted using ViT B, the cross backbone generalization capability needs to be checked. The authors are encouraged to replace the backbone with some CNN architectures to see its performance (e.g., ResNet50).
>
> Thanks for the feedback! To address this, we have added additional experiments for our method using the suggested CNN architecture (i.e., ResNet50). See Appendix E.2
>
> > Per-domain performances are not provided, which are also interesting.
>
> Thanks for the feedback! To address this, we have added per-domain performance results for a more comprehensive understanding in Appendix D.3.
>
> ### **Add Algorithm Format for Clarity**
> > Section 4.2 needs more details to make the method clearer. The authors are encouraged to describe it via algorithm format.
>
> Thanks for the feedback! To address this point, we have updated the draft to also present our method in algorithm format in Section 4.
>
> ### **Other Clarification Points**
> > The number of expert models grows linearly with the number of domains, potentially introducing significant computational and memory overhead in large-scale settings.
>
> Thank you for pointing out this limitation! We agree that the number of expert models scaling linearly with the number of domains can lead to increased computational and memory costs, especially in large-scale settings. We have acknowledged this point in the Limitations section (Section 7). We have also updated the draft to include a more explicit discussion of the computational and memory requirements of our method in Section 7 (and Appendix B.3).
>
> > The study primarily targets classification problems; the applicability of G2D to other task types (e.g., regression, structured prediction) remains unexplored.
>
> Thank you for noting this limitation! You’re right that our study focuses on classification tasks, and the applicability of G2D to other settings such as regression or structured prediction remains an open question. To clarify this point and highlight it as a promising direction for future work, we have updated the discussion in Section 7 to reflect this.
>
> Thanks again for your thoughtful review. We hope this addresses your concerns!

---

> > ### Comment · Reviewer_RphK · 2025-05-26
> > **Response**
> >
> > Thanks the authors for the detailed response, my concerns are mostly solved.

---

### Review · Reviewer_mhFX · 2025-04-10

**Summary Of Contributions:**

1. This paper proposes Generate to Discriminate (G2D), a domain-incremental learning method that leverages synthetic data to train a domain-discriminator that routes samples at inference time to the appropriate expert. Combined with a group of models trained with different domain data, G2D could be used in scenarios that require domain-incremental learning. Experimental results show that G2D outperforms several domain-incremental learning methods in both vision and language modalities.
2. The paper conducts careful analysis to understand the domain discrimination boundaries of G2D. While leveraging synthetic data is not a new idea, one contribution of the paper is the empirical evidence that using synthetic data to train a domain discriminator is more effective than using the same synthetic samples to train the downstream label classifier.

**Audience:**

Yes

**Broader Impact Concerns:**

I do not see broader impact concerns in this work.

**Claims And Evidence:**

Yes

**Requested Changes:**

1. Question: How many generators the method need to keep if there are $N$ domains?
2. Following up with the question, I think the paper could be strengthened if there are more discussions about factors besides the performance. For example, compared with using synthetic data to train downstream classifiers, the proposed method may require more training or storage budget.

**Strengths And Weaknesses:**

Strengths:
1. The paper proposes G2D, an effective domain-incremental method that leverages synthetic data to train a domain-discriminator that routes samples at inference time. The proposed method outperforms several previous methods, especially on DomainNet.
2. The benchmark introduced by this work is more realistic than previous datasets and would be useful for future study of domain-incremental learning.

Weaknesses:
1. Even though the paper conducts some analysis and visualization in Section 6.1, it lacks more in-depth understanding of why using synthetic data to train domain-discriminator would lead to much stronger performance than using synthetic data to train downstream classifiers directly.

---

> ### Author Response · Authors · 2025-05-21
> **Response to Reviewer mhFX**
>
> Dear Reviewer mhFX,
>
> Thank you for the feedback! We’re grateful for your acknowledgement of the value of our work. We address your comments below.
>
> ### **Clarification Points**
>
> > Question: How many generators the method need to keep if there are N domains?
>
> Thanks for the question! Our method does not require keeping generators from previous domains. As a result, we only need to keep **one generator** at any given time (i.e., At the Nth domain, we only require the current domain’s generator). We have clarified this point in the updated draft (see Section 4.1).
>
> > Following up with the question, I think the paper could be strengthened if there are more discussions about factors besides the performance. For example, compared with using synthetic data to train downstream classifiers, the proposed method may require more training or storage budget.
>
> Thanks for the feedback! To address this point, we have added a new subsection in Section 7 (and additional details in Appendix B.3) that explicitly discusses the computational and storage requirements of our method. We hope this provides further guidance for practitioners when choosing between approaches based on their available resources.
>
> ### **Understanding of empirical findings**
>
> > Even though the paper conducts some analysis and visualization in Section 6.1, it lacks more in-depth understanding of why using synthetic data to train domain-discriminator would lead to much stronger performance than using synthetic data to train downstream classifiers directly.
>
> Thanks for the feedback! To address this, we have revised the discussion in Section 6 to articulate our hypothesis and empirical justification in greater detail. In short, we attribute the observed performance gap to the fact that domain discrimination primarily relies on learning higher-level differences (i.e., global features in style or structure)—such as color palette, texture, or linguistic patterns. These higher-level signals are more likely to be preserved (than lower-level signals), even when the generated samples contain artifacts or imperfections from the generative model. This aligns with prior observations that have been documented for the language domain, where generative models have been shown to cluster by domain without supervision [1]. In Section 6.1, we empirically support this claim by showing that domain-discrimination performance closely matches oracle routing accuracy, suggesting that generative models are indeed effective at capturing domain-level variation. In contrast, downstream label classification often depends on fine-grained, class-specific details that are more sensitive to generative artifacts. While a full theoretical analysis is beyond the scope of this work, we highlight this in the updated draft as an important direction for future research.
>
> [1] Aharoni, et. al. Unsupervised Domain Clusters in Pretrained Language Models.
>
> Thanks again for your thoughtful review. We hope this addresses your concerns!

---

> > ### Comment · Reviewer_mhFX · 2025-05-24
> >
> > I think the revised version does a great job in clarifying my previous questions. I thank the authors for the careful response.

---

### Review · Reviewer_3i76 · 2025-05-06

**Summary Of Contributions:**

This paper presents Generate to Discriminate (G2D), a method that learns a domain discriminator for routing during domain-incremental learning. The proposed discriminator helps to decide which expert classifier to route an example to at inference time. The discriminator is trained on synthetic data from previous tasks. The authors claim that training this discriminator on synthetic data provides almost the same performance than training it on actual task data. Experiments are presented for standard domain-incremental benchmarks, and the proposed method's performance is compared against several baselines including MTL, regularization-based methods, experience replay methods, and diverse oracles.

**Audience:**

Yes

**Broader Impact Concerns:**

No.

**Claims And Evidence:**

Yes

**Requested Changes:**

- Please examine additional continual learning metrics, larger number of tasks, and provide conclusions of the behaviour of the proposed method.
- Please provide a more comprehensive analysis of the computational cost of the proposed method.
- Please examine the impact of the chosen expert classifiers on the performance of the system.

**Strengths And Weaknesses:**

Strengths:
- The paper studies an important problem of domain-incremental learning, in particular an expert routing mechanism. The proposed method seems valuable as it focuses on the problem of training/updating the discriminator.
- The paper is well-writen and is easy to follow. Technical aspects, when presented, are presented to the right level of detail.
- The paper is accompanied by a new benchmark on a specific type of task which adds value to the proposed method.

Weaknesses:
- As a continual learning paper, it is expected that the experiments examine important aspects such as forgetting and/or transfer (forward or backward). The current set of experiments is limited to examining overall accuracy, which can be misleading at times. Furthermore, what is the performance across the sequential training/inference? What is the performance while the number of tasks increases? All these aspects are fundamental in continual learning research.
- Despite some numbers presented in the appendix, the paper lacks a serious examination of the computational requirements of the proposed method, especially in comparison to other methods. While a clear advantage is the lack of need of storing past task examples, what is the cost of generating examples for tasks especially as the number of tasks/domains increases? Some theoretical bounds of thorough experimental examination of this would be necessary to convince the reader that the method does not incur significant extra-costs compared to other strategies.
- A single architecture is chosen for expert classifiers in the vision and text domains. I would expect experiments to show that the discriminator works with at least a few types/architectures as expert classifiers, in terms of both performance and computational cost.

---

> ### Author Response · Authors · 2025-05-21
> **Response to Reviewer 3i76**
>
> Thank you for the feedback! We’re grateful for your acknowledgement of the value of our work. We address your comments below.
>
> ### **Additional Metrics**
>
> > As a continual learning paper, it is expected that the experiments examine important aspects such as forgetting and/or transfer (forward or backward). The current set of experiments is limited to examining overall accuracy, which can be misleading at times. Furthermore, what is the performance across the sequential training/inference? What is the performance while the number of tasks increases? All these aspects are fundamental in continual learning research. Please examine additional continual learning metrics and provide conclusions of the behaviour of the proposed method.
>
> Thanks for the feedback! To address this point, we additionally report (1) Forgetting, as a measure of *backward transfer* (i.e., how does learning on domain $t$ influences the performance of previous domains $s, 1 \leq s < t$) [1]; and (2) Learning Accuracy, as a measure of *forward transfer* (i.e., the learning capability when the model encounters a new domain $t$) [2]. In short, we observe that our method leads to less forgetting while retaining high learning accuracy (see Appendix D.1). Additionally, we also report the performance across sequential training to understand how performance evolves while the number of tasks increases (see Appendix D.2).
>
> >Larger number of tasks
>
> Thanks for this suggestion! We would like to clarify that CORe50 (one of the datasets that we investigate) consists of the longest domain sequence out of existing (and publicly available) domain incremental learning benchmarks. Due to this, it is difficult for us to investigate longer domain sequences at this current point in time. Yet, we understand that a more large-scale evaluation will indeed be an interesting additional experiment (as new benchmarks become available that can support this) and have acknowledged this point in the Discussion and Limitations section (see Section 7).
>
> ### **Computational Cost and Storage Requirements**
> > Please provide a more comprehensive analysis of the computational cost of the proposed method.  Despite some numbers presented in the appendix, the paper lacks a serious examination of the computational requirements of the proposed method, especially in comparison to other methods. While a clear advantage is the lack of need of storing past task examples, what is the cost of generating examples for tasks especially as the number of tasks/domains increases? Some theoretical bounds of thorough experimental examination of this would be necessary to convince the reader that the method does not incur significant extra-costs compared to other strategies.
>
> Thanks for the feedback! We have added a new subsection in Section 7 (and Appendix B.3) that explicitly discusses the computational and storage requirements of our method in comparison to other approaches. This includes clarifying the cost of generating synthetic samples as the number of tasks/domains increases. We believe that these additional compute and memory requirements are relatively manageable compared to recent practice in machine learning pipelines. We hope this addition provides more transparency and further guidance for practitioners when choosing between approaches based on their available resources.
>
> ### **Additional Experiments with different architecture**
> > Please examine the impact of the chosen expert classifiers on the performance of the system. A single architecture is chosen for expert classifiers in the vision and text domains. I would expect experiments to show that the discriminator works with at least a few types/architectures as expert classifiers, in terms of both performance and computational cost.
>
> Thanks for the feedback! To address this point, we have added additional experiments using a CNN architecture (i.e., ResNet50 backbone) for our expert classifiers and domain discriminator (See Appendix E.2). We note that the computational cost requirements are comparable to those required by the ViT-B16 backbone (See Appendix B.3).
>
> [1] Lopez-Paz and Ranzato. Gradient episodic memory for continual learning.
> [2] Riemer et al. Learning to learn without forgetting by maximizing transfer and minimizing interference.
>
> Thanks again for your thoughtful review. We hope this addresses your concerns!

---

> > ### Comment · Reviewer_3i76 · 2025-05-26
> > **Thank you**
> >
> > I appreciate the efforts of the authors to address my concerns/answer my questions.

---

### Decision · Action_Editor_cZYx · 2025-07-02

**Recommendation:** Accept as is

**Audience:**

Yes

**Audience Explanation:**

All reviewers agree this work is relevant to the continual learning community.

**Claims And Evidence:**

Yes

**Claims Explanation:**

This article introduces a novel domain-incremental framework that uses synthetic data to train a domain discriminator for routing to the appropriate expert at inference time, rather than using such data directly for label classification. The reviewers generally agree that the method is well-motivated and technically sound, with strong empirical effectiveness across both vision and language tasks. In particular, the use of synthetic data for domain routing, coupled with strong performance without storing prior real data, was seen as a valuable contribution to continual learning, especially for domains where privacy is an issue. The initial reviews raised concerns around evaluation depth (evaluation, transfer), computational costs, generality, and clarity of some aspects that were addressed via substantial revisions from the authors, including new metrics, broader analysis, additional experiments, and improved presentation. The reviews did note that the paper could still be improved by a deeper theoretical understanding of the method, and by better addressing scalability limitations with the growing numbers of models, but reviewers were otherwise satisfied with the revisions and unanimously recommend acceptance.